# Putting ICU triage guidelines into practice: A simulation study using observations and interviews

Inger L. Abma[1]*, Gert J. Olthuis[1], Irma T. H. M. Maassen[2], Marjan L. Knippenberg[2], Miriam Moviat[3], Annie J. Hasker[4], A. G. Buenen[5¤], Bernard G. Fikkers[6], Anke J. M. Oerlemans[1]

**1** IQ healthcare, section Ethics of healthcare, Radboud Institute of Health Sciences, Radboud University Medical Center, Nijmegen, The Netherlands, **2** IQ healthcare, Radboud Institute of Health Sciences, Radboud University Medical Center, Nijmegen, The Netherlands, **3** Department of Intensive Care Medicine, Jeroen Bosch Hospital, 's-Hertogenbosch, The Netherlands, **4** Pastoral Care Department, Isala, Zwolle, The Netherlands, **5** Department of Emergency Medicine, Bernhoven Hospital, Uden, The Netherlands, **6** Department of Critical Care Medicine, Radboud University Medical Center, Nijmegen, The Netherlands

¤ Current address: Department of Emergency Medicine, Maxima MC, Veldhoven, The Netherlands
* inger.abma@radboudumc.nl

**Data Availability Statement:** Participants have given their informed consent for the use of anonymized fragments of qualitative data. Participants did not consent to provision of the full

## Abstract

### Background

The COVID-19 pandemic has prompted many countries to formulate guidelines on how to deal with a worst-case scenario in which the number of patients needing intensive care unit (ICU) care exceeds the number of available beds. This study aims to explore the experiences of triage teams when triaging fictitious patients with the Dutch triage guidelines. It provides an overview of the factors that influence decision-making when performing ICU triage with triage guidelines.

### Methods

Eight triage teams from four hospitals were given files of fictitious patients needing intensive care and instructed to triage these patients. Sessions were observed and audio-recorded. Four focus group interviews with triage team members were held to reflect on the sessions and the Dutch guidelines. The results were analyzed by inductive content analysis.

### Results

The Dutch triage guidelines were the main basis for making triage decisions. However, some teams also allowed their own considerations (outside of the guidelines) to play a role when making triage decisions, for example to help avoid using non-medical criteria such as prioritization based on age group. Group processes also played a role in decision-making: triage choices can be influenced by the triagists' opinion on the guidelines and the carefulness with which they are applied. Intensivists, being most experienced in prognostication of critical illness, often had the most decisive role during triage sessions.

raw dataset to persons other than the research team. Since the raw interviews and transcripts contain sensitive information, even anonymized raw data can compromise their confidentiality. Therefore, current Dutch privacy law and institutional regulations prevent the a priori sharing of the full raw dataset. Considering the importance of data-sharing and providing insight into the research, data access will be considered upon request, evaluating each inquiry individually. Requests for data access may be sent to the department of IQ healthcare of the Radboudumc at iqhealthcare@radboudumc.nl.

**Funding:** This research was funded by a ZonMw grant (project number 10430022010004, https://www.zonmw.nl/en/) The grant was not awarded to a specific author. The funders had no role in study design, data collection and analysis, decision to publish, or preparation of the manuscript.

**Competing interests:** The authors have declared that no competing interests exist.

## Conclusions

Using the Dutch triage guidelines is feasible, but there were some inconsistencies in prioritization between teams that may be undesirable. ICU triage guideline writers should consider which aspects of their criteria might, when applied in practice, lead to inconsistencies or ethically questionable prioritization of patients. Practical training of triage team members in applying the guidelines, including explanation of the rationale underlying the triage criteria, might improve the willingness and ability of triage teams to follow the guidelines closely.

## Introduction

The coronavirus disease 2019 (COVID-19) pandemic has, since its start in early 2020, overwhelmed healthcare systems all over the world [1–4]. Many patients with severe COVID-19 needed to be admitted to the intensive care unit (ICU): a hospital ward that can provide immediate life-saving care to patients who have, or are at risk of, life-threatening organ dysfunction, and which is staffed by specialized healthcare professionals. The primary goal of intensive care is to support organ function and prevent further physiological deterioration, while the underlying disease is treated [5]. ICU care may be needed after an accident, major surgery, or due to a severe illness such as COVID-19. Patients with COVID-19 may be admitted to the ICU if they can no longer sufficiently oxygenate their body and therefore need to be placed on a ventilator. The mortality rate of COVID-19 patients that have been admitted to the ICU is approximately 24% [6], with a higher mortality rate of approximately 40% found in patients >70 years of age [7]. Patients who recover will on average have spent two weeks on a ventilator [6]. A long stay in the ICU takes a heavy toll on a patient's physical and mental health and ICU survivors face a long recovery and reduced quality of life [8]. Especially in the elderly and vulnerable, it should always be carefully considered whether an ICU admission is suitable and desirable for the patient [9].

Due to extreme demand for ICU care during the COVID-19 pandemic, many countries faced acute shortages of ICU staff, beds and ventilators. A worst-case scenario came close to reality: a situation in which patients would need to be refused ICU admission because no more beds are available. Numerous guidelines were developed all over the world for this type of scenario: how should healthcare professionals choose which patient ought to receive an ICU bed in times of shortage?

Several international comparisons of triage guidelines have been published [10–14]. The main ground for triage in all studied triage guidelines is "maximizing benefit", meaning a maximum number of lives saved and/or life years saved, sometimes also suggesting quality of life should be taken into account. However, not all guidelines clearly operationalize how healthcare professionals should make decisions based on these criteria [10]. The Dutch guidelines are one of the exceptions: they describe a step-by-step ICU triage process which aims to be practically applicable [10, 15, 16]. If ICU triage becomes necessary in the Netherlands, following the triage guidelines is legally required. It is therefore highly important that triage teams are willing and able to use these guidelines as intended.

To our best knowledge, practical testing of triage guidelines in a scientific study has so far not taken place. This study aims to explore the experiences of triage teams when triaging fictitious patients with the Dutch triage guidelines. This provides insight into the acceptability and applicability of the Dutch guidelines. Moreover, this study aims to answer a broader question: which factors influence decision-making when performing ICU triage with triage guidelines?

Answering these questions will benefit an international audience of triage teams, writers of ICU triage guidelines, and hospitals adapting national guidelines to local protocols.

## Methods

### Design

In this qualitative study eight triage teams from four Dutch hospitals were presented with files of fictitious patients in need of an ICU bed. The teams were provided with the Dutch ICU triage guidelines and instructed to triage these patients in the way they would during a crisis situation. Focus group interviews with triage team members from the four hospitals were then held to reflect on the sessions and the Dutch guidelines. Data collection took place in March-May 2021.

The researchers who collected and analyzed the data are experienced in conducting qualitative research in the medical field but are not medical professionals. The researchers did not know the participants before the study started.

### Ethics approval and consent to participate

This type of study does not fall under the Dutch "Medical Research Involving Human Subjects Act (WMO)" (https://english.ccmo.nl/), therefore ethical approval is not required. A written confirmation that the WMO does not apply to this study was obtained from the local Medical Ethics Committee 'CMO Regio Arnhem-Nijmegen' (dossier number 2020–7152). All participants were sent participant information and signed written informed consent for participation in the study and use of the acquired data for publication.

### Dutch ICU triage guidelines

The Dutch ICU triage guidelines were published during the COVID-19 pandemic in two complementary documents: one document with medical triage criteria (to be applied first) [15]; and a second document with non-medical triage criteria [16]. These documents were written by Dutch physician organizations (Federation of Medical specialists, Dutch Association for Intensive Care, the Royal Dutch Medical Association). Ethicists and the Dutch Patient Federation were involved as advisors, and for the second document citizen groups were also consulted. The ICU triage guidelines are to be applied only in a national situation in which the Dutch Ministry of Health has declared that "phase three" goes into effect, i.e. there is a severe shortage of ICU beds all across the country and ICU triage is necessary. Following the Dutch triage guidelines during phase three is legally required: patients have to be triaged according to the presented step-by-step plan. Patients who are already in the ICU are never triaged; they can stay in the ICU as long as they still have a reasonable chance of survival.

The step-by-step triage process of the medical and non-medical Dutch ICU triage guidelines is shown in Fig 1. The main principle underlying the first four steps of the Dutch guidelines is maximizing benefit, in this case saving the most lives. The first step consists of a set of in- and exclusion criteria to be applied to an individual patient irrespective of whether there are other patients waiting for an ICU bed at that moment. If only one patient is left after applying these criteria, this patient is admitted. If there are multiple patients left in need of an ICU bed, a stricter exclusion based on frailty will be applied in step 2, and if necessary in step 3 the remaining patients will be compared on chance of survival. If after these steps there are still more patients in need of ICU admission than there are beds, the non-medical criteria will be applied. At step 5, priority is given to healthcare professionals who worked with COVID-19 patients and contracted COVID-19 when there was regional scarcity of personal protection equipment. At step 6, patients in a younger age group are prioritized. If the number of patients

> **General ICU triage rules**
>
> - Patients should only be referred for triage after balancing the pros and cons of ICU admission for the individual patient (just like in a regular situation)
> - Patients who are already in the ICU are not triaged, but undergo daily evaluation for reasonable chance of survival
> - Discrimination based on sex, ethnicity, nationality, societal position, physical or cognitive or physical impairments, personal relationships, ability to pay or juridical status is not allowed
> - Whether the illness can potentially be considered someone's "own fault" should not be taken into account
>
> ---
>
> **Medical criteria**
>
> Step 1: Stricter inclusion criteria (i.e. is ICU care strictly necessary?) and exclusion criteria for ICU admission (e.g. short life expectancy due to (co)morbidity or due to severe frailty (CFS ≥6[c]))
>
> Step 2: CFS ≥5[c] or needing ECLS = no ICU admission;
>
> Step 3: Comparison between patients: patients with >20% more chance of survival of the ICU-admission than other patients have priority
>
> ↓
>
> ---
>
> **Non-medical criteria**
>
> Step 4: Comparison between patients: patients who are expected to have shorter length of stay than other patients have priority
>
> Step 5: Priority for healthcare professionals with COVID-19 who worked in the care for patients with COVID-19 during a time period in which there was a regional scarcity of personal protection equipment
>
> Step 6: Comparison between patients: patients in a younger age category have priority. The categories are 0-20, 20-40, 40-60 etc.
>
> Step 7: Drawing lots

**Fig 1. Overview of Dutch triage guidelines [15, 16][a,b].** CFS = clinical frailty scale; ECLS = extra corporeal life support; ICU = intensive care unit a. Each subsequent step is only relevant for a given triage situation if, after the previous step(s), there are still more patients in need of ICU admission than there are beds. E.g. if, at step 4, two patients both have the same chance of survival (greater than potential other patients) they will both be taken into consideration at step 5, etc. b. In addition to phase three exclusion criteria, regional crisis teams can decide that elective surgeries should be cancelled when the number of regional ICU beds starts to become constrained. C. For patients ≥ 65 years old; for adult patients <65 this criterion may only be used if this is relevant for the prognosis in the ICU (i.e. medical characteristics/disabilities that do not impact prognosis should not be taken into account).

still exceeds the number of available ICU beds, then chance (drawing lots) is to be the deciding factor. Steps 3–7 are to be applied by a triage team consisting of multiple healthcare professionals, including at least one intensivist. Steps 1–2 can be applied by a team or an intensivist by themselves. Triage teams do not communicate directly with patients and their families.

## Participants

When this study started recruitment, the four participating hospitals (Table 1) had prepared for a potential "phase three" situation by recruiting healthcare professionals for triage teams. These persons were approached for participation in this study. One hospital included up to six

Table 1. Characteristics of the participating hospitals.

| | Ranked from smallest to largest ICU department[a] | | | |
|---|---|---|---|---|
| University hospital y/n | No | No | Yes | No |
| Nr of ICU beds in normal situation | 8 | 14 | 26 | 30 |
| Nr of beds in hospital | 220 | 630 | 1065 | 620 |

a. In order to keep the quotes in the results section anonymous, the hospitals are not given a number in the methods section

members in its teams: three physicians, one or two medium or intensive care nurses, and one medical psychologist, ethicist or spiritual caregiver. The standard teams of the three other hospitals consisted of three or four physicians, including at least one intensivist. All triage team members had received some internal training in their hospital in how to use the Dutch guidelines for ICU triage prior to the simulated triage session.

For the focus groups, participants of the triage sessions were selected based on willingness to participate and availability. Four online focus group interviews were held, each with in total four or five participants from at least three different hospitals.

## The triage sessions

Sixteen fictitious patient records of patients needing ICU care ("cases") were used in this study: eight cases for the first team in each hospital (session A), and eight cases for the second team (session B). The cases were based on suggestions of healthcare professionals who worked in the different participating hospitals but did not take part in the simulated triage sessions. They were asked to suggest a combination of COVID-19 and non-COVID-19 cases that might be challenging to triage. More cases were added that were aimed at ensuring that all different criteria in the guidelines could potentially be used during the session. The fictitious patient records of the cases were then written by medical professionals of the participating hospitals. The records contained the medical history, current disease progression for which ICU admission is needed and some social information. Most records were 350–500 words. The cases of patients with COVID-19 were slightly adjusted for one hospital as it offered an additional treatment (Optiflow) outside of the ICU, meaning the care trajectory in the hospital would be slightly different. A brief description of the sixteen cases can be found in Table 2. One pilot triage session with the cases of session A was held in one of the hospitals with a triage team that did not participate in the study. This confirmed that the envisioned set-up of the fictional triage sessions was feasible in terms of time and provided information.

Each team was asked to prioritize eight cases, with the instruction: which case would you admit if only one bed was available? Who if only two beds were available? Etc. This approach was chosen so that each case would be compared to several others. Teams could also decide to refuse ICU care. The hypothetical date for triage was 21 February 2021. At this time, in the Netherlands vaccines were only available for certain high-risk groups and point-of-care testing for COVID-19 was not yet possible. The government had set rules in order to limit physical meetings among citizens: people were encouraged to work from home if possible, there was a curfew in place from 9PM until 4:30AM, and it was strongly advised not to have more than one visitor per day. At the start of each session, each triage team member was presented with the eight cases on paper and was given a hardcopy of the Dutch triage guidelines. No specific instructions were given with regard to how to use the guidelines. Each session (including introduction and instructions) could take a maximum of 1.5 hours. After the instructions were given, the researchers remained present but were not involved in the session.

**Table 2. Code and brief description of the cases.**

| **Cases for first team of each hospital** | |
|---|---|
| **COVID-19 patients** | |
| Code | Description |
| C1 | 54-year-old man with obesity, diabetes mellitus and hypertension, works in the hospital IT department |
| C2 | 66-year-old woman with diabetes mellitus and some renal insufficiency, originally from Morocco, her children insist on ICU admission |
| C3 | 40-year-old woman in her third month of pregnancy after IVF, no comorbidity |
| C4 | 61-year-old man with Parkinson's disease and an advance euthanasia directive in case of severe Parkinson, not discussed with patient if he wants ICU admission, currently unresponsive |
| **Regular (non-COVID-19) patients** | |
| R1 | 17-year-old woman, 8[th] suicide attempt (intoxication) |
| R2 | 31-year-old woman who recently gave birth by emergency caesarian section, HELPP syndrome |
| R3 | 37-year old woman, mother of three young children, suffering from pneumonia and metastasized ovarian cancer |
| R4 | 60-year old man with pneumonia and HIV with a rising viral load |
| **Cases for second team of each hospital** | |
| **COVID-19 patients** | |
| Code | Description |
| C5 | 40-year-old man with Down syndrome, otherwise healthy |
| C6 | 64-year-old man, intensivist at the hospital of the triage team, overweight and with hypertension |
| C7 | 21-year-old woman, student, overweight, was infected with COVID-19 at an illegal party |
| C8 | 63-year old woman, no comorbidity, babysits her grandchildren |
| C9 | 41-year-old woman with pneumonia or COVID-19 (test results not yet received), found in a park, looking neglected |
| **Regular (non-COVID-19) patients** | |
| R5 | 22-year-old woman, 8[th] suicide attempt (intoxication) |
| R6 | 50-year-old man with several neurological trauma after a traffic accident |
| R7 | 54-year-old man with an abdominal aneurysm requiring surgery in 3–7 days |

HELLP syndrome = "Hemolysis, Elevated Liver enzymes, and Low Platelet count" syndrome; HIV = human immunodeficiency virus; ICU = intensive care unit; IT = information technology; IVF = in vitro fertilization

The triage sessions were audio recorded and transcribed verbatim. Furthermore, the sessions were observed by the researchers based on an observation guide focusing on group dynamics/interactions, process of decision-making, body language, emotions of the participants and general atmosphere.

## Focus group interviews

Four online focus groups were held with a mix of professionals from the different participating hospitals. Beforehand, all participants were sent an (anonymous) overview of the decisions and arguments of the teams of all participating hospitals who triaged the same cases. During the focus group, participants were asked to reflect on this overview, their triage session and the choices made, as well as on the Dutch triage guidelines and the composition of the triage teams.

## Analysis

An inductive content analysis was performed using the constant comparative method, in which no pre-defined codebook or hypothesis is used, but codes and categories are constructed from the data through an iterative process. Codes were assigned to the transcripts and

observation forms, and constantly adjusted where needed based on new information gathered from newly coded transcripts. Codes were then sorted into categories.

Two rounds of coding and categorization took place. First, a round with very broad coding for relevant quotes and observations regarding the content and process of triage including detailed codes of arguments per case, which resulted in a first codebook with categories. The second coding round was focused on the specific research question "Which factors influence triage decisions?" in which the transcripts were all coded again and codes were merged, added, removed and adjusted where needed and categories were rephrased to suit this specific research question. This resulted in one final codebook with categories for both the triage sessions and the focus groups. During both coding rounds, all documents were first coded by IA, after which IM or MK critically reviewed the assigned codes and adjusted or added codes where they thought this was necessary. Frequent meetings were held to discuss the coding and reach consensus. Between the first and second coding rounds, and halfway through the second coding round, the coding team discussed the codes and categories with AO and GO and adjustments were made based on consensus.

## Results

### General observations

Characteristics of the participants of the triage sessions (n = 30) and the focus group interviews (n = 17) are shown in Table 3. Average length of the triage sessions (minus instructions) was 56 minutes (range 37–76 minutes). Teams were aware that there was enough time available so they could have discussions when needed. All sessions had an informal atmosphere: the professionals used each other's first names and lightly joked around. Despite having received some training, many triage team members were not familiar with the triage guidelines in detail: many sessions contained moments of confusion and team members correcting each other's interpretations of the criteria. The triage decisions were made based on consensus. Prioritization of patients was similar among the teams, though not exactly the same (Table 4; see Table 2 for case codes). Argumentation regarding why a patient was prioritized in a certain way (i.e. at which step of the triage guidelines the decision was made and why) differed more frequently among teams. Some differences in prioritization were influenced by differences in the medical-technical capabilities of the hospital, such as whether Optiflow treatment was only available in the ICU or not.

### Applicability and acceptability of the Dutch triage guidelines

The Dutch triage guidelines were considered highly applicable, with only minor points of criticism regarding the clarity. For example, if a patient is 40 years old, are they in the age category

**Table 3. Characteristics of the participants.**

|  | Triage sessions (n = 30) | Focus groups (n = 17) |
|---|---|---|
| Age (average, range) | 50.6 (34–68) | 52.4 (37–68) |
| % Female | 50% | 35% |
| Years of experience in profession (average, range) | 17.1 (1.5–42) | 18.4 (4–42) |
| Physician (n) | 25 | 14 |
| Intensivist (n) | 9 | 6 |
| Other[a] (n) | 16 | 8 |
| Nurse (n) | 3 | 2 |
| Medical psychologist/spiritual caregiver/ethicist (n) | 2 | 1 |

a. This category includes oncologists, cardiologists, nephrologists and surgeons, amongst others.

**Table 4. Triage decisions per team[a].**

| Teams session A | | | | |
|---|---|---|---|---|
| **Bed** | **Hospital 1[b]** | **Hospital 2** | **Hospital 3** | **Hospital 4** |
| 1 | R1 | R1 | R1 | R1 |
| 2 | R2 | R2 | R2 | R2 |
| 3 | C3[c] | C3 | C3 | C3 |
| 4 | C1[c] | C1 | C1 | C1 |
| 5 | C2[c] | C2 | C2[c] | C2 |
| 6 | C4[c] | C4[d] | C4[c] | R4[d] |
| 7 | Not admitted: | R4[d] | Not admitted: | Not admitted: |
| 8 | R3[e], R4[f] | Not admitted: R3[g] | R3[e], R4[f] | R3[e], C4[h] |

| Teams session B | | | | |
|---|---|---|---|---|
| **Bed** | **Hospital 1[b]** | **Hospital 2** | **Hospital 3** | **Hospital 4** |
| 1 | R5 | C7 | R5 | C7 |
| 2 | C7 | C8 | C7 | C8 |
| 3 | C5 | C5 | C8 | C6 |
| 4 | Not admitted: C6[i], C8[i], C9[i], R6[j], R7[k] | C6 | Not admitted: C5[l], C6[i], C9[i], R6[j], R7[k] | C5 |
| 5 | | C9 | | Not admitted: C9[i], R5[m], R6[j], R7[k] |
| 6 | | R5 | | |
| 7 | | Not admitted: R6[j], R7[k] | | |
| 8 | | | | |

a. An overview of codes and their corresponding cases can be found in Table 2.

b. The COVID-19 cases were slightly adjusted for Hospital 1 as it offered an additional treatment (Optiflow) outside of the ICU. Some saturation parameters were therefore different than those of the other hospitals.c. Lots were drawn for the cases sharing a table cell.

d. Case would only be admitted to the ICU if, after a conversation with the patient about what this might mean for their future well-being, they want to be admitted

e. Argument for exclusion: does not meet inclusion criteria (short life expectancy due to malignancy)

f. Argument for exclusion: exclusion criterion met (clinical frailty scale 6)

g. Argument for exclusion: short life expectancy (own consideration, not based on guidelines)

h. Argument for exclusion: patient likely does not want to be admitted considering the advance euthanasia directive

i. Argument for exclusion: does not meet inclusion criteria (saturation not sufficiently low and no exhaustion)

j. Argument for exclusion: exclusion criterion met (severe neurological damage)

k. Argument for exclusion: no elective surgeries during phase three

l. Argument for exclusion: does not meet inclusion criteria (saturation not sufficiently low; work of breathing/exhaustion was not taken into account)

m: Argument for exclusion: need for ICU admission not high enough (not based on guidelines)

"20–40" or "40–60"? It was generally appreciated how the guidelines help guide and document the triage decisions. Some professionals expressed appreciation for how the guidelines help teams stay objective.

> *Intensivist*: *I feel like it provides a very clear guideline, because, for example, take the lady with those repeated suicide attempts, her case makes everybody feel a lot of resistance. And I do think that all groups have that discussion. . . when it comes to emotion. But, you can actually completely remove that emotion, especially if you can look at it objectively. . . Is someone still treatable?* (Focus group 2)

The content of the guidelines was generally considered acceptable, with several professionals expressing praise for the amount of thought and effort that went into developing them. However, not all professionals agreed with all aspects of the guidelines. For example, several professionals pleaded for stricter exclusion criteria. And while none of the participants argued

against using age if no medical difference was obvious, some felt that the age categories were too broad, or that they were unjust in their application because a very small age difference could result in prioritizing one patient over another (e.g. someone aged 39 would be prioritized over someone aged 40). Lastly, during the sessions it became clear that some situations which some professionals considered intuitively relevant for triage were not addressed in the guidelines, such as pregnancy and severe depression.

## Factors that influence ICU triage decision-making with triage guidelines

The factors that influence triage decisions fall in two main themes: *arguments for prioritization* and *group processes*. These are explained in more detail below. Table 5 gives an overview of the categories and factors, which are explained in more detail below.

**Arguments for prioritization.** *Medical assessments in the guidelines.* Most of the teams used the triage guidelines as their main basis for the triage decisions. For several criteria it is necessary to perform a medical assessment, for example: what is a patient's chance of survival and how many days in the ICU will they need? The estimation of whether or not two patients differed 20% in chance of survival regularly differed between teams. Intensivists had the most experience and expertise in performing assessments in critically ill patients, though several of them indicated that they found it difficult to predict what would happen to an individual patient. The assessment of the intensivists were usually decisive.

**Table 5. Overview of categories and factors.**

| Category | Factor | Topics |
|---|---|---|
| Arguments for prioritization | Medical assessments in the guidelines | • assessments often differ between teams<br>• assessment of intensivist decisive<br>• impact of cognitive capabilities on time in ICU |
| | Unease at applying non-medical criteria of the guidelines | • unease at using age and drawing lots<br>• important to draw lots when there is uncertainty |
| | The team's own considerations: content | Considerations for individual patients<br>• ICU medically necessary and beneficial?<br>• wish of patient to be admitted<br>Current mental state<br>• emotional well-being of patient<br>Situation after ICU discharge<br>• rehabilitation/quality of life after ICU admission<br>• low life expectancy due to psychiatric illness<br>Social context (in practice not taken into account)<br>• for example health behavior, parenthood<br>Other<br>• pregnancy<br>• unease regarding COVID patients always taking priority over elective surgery patients |
| | The team's own considerations: role | • arguments completely unrelated to guidelines should not be used<br>• own considerations sometimes "translated" into criteria of guidelines<br>• important to follow guidelines strictly |
| Group processes | Opinion on the use of guidelines for triage | • use of guidelines during a crisis not feasible |
| | Careful application of the guidelines | • criteria sometimes skipped or not applied exactly as written down<br>• team members help and correct each other |
| | Handling of missing information | • CFS score missing: score is based on available information |
| | The role of intensivists versus non-intensivists | • large role of intensivist due to most relevant experience<br>• important that intensivist listens to other team members<br>• potentially difficult for non-intensivists to be equal partner |

CFS = clinical frailty score; ICU = intensive care unit

*Intensivist: I don't think that the 20% difference is applicable, [. . .] [C4] is still fairly vital and he doesn't have a Parkinson's crisis right now or anything like that. In terms of rehabilitation, you can argue about that, his chances are really worse. But is that 20%? I can't judge that.*

*Non-intensivist: [case C2] has obesity, I currently think she has been dealt the worst hand, but I don't have much experience with Parkinson's.*

*Intensivist: I don't think it is possible to make that 20% difference.*

*Non-Intensivist: So we'd draw lots for spot 5 and 6. Right. Do we agree*? (Hospital 3, session A)

Most triage team members did not consider the decisive role of the intensivist in these estimations a problem. However, one physician (non-intensivist) worried that much of the triage process was in practice based on one person's estimation.

*Non-intensivist: You start wondering*: "*What would another intensivist think about that*? *What would another hospital think about that*?" *(Focus group 3)*

Regarding estimating expected length of ICU stay, in addition to medical factors, cognitive capabilities were frequently considered a risk factor for a longer length of stay due to prolonged weaning. This was relevant in the case of the patient with Down syndrome (C5), which two teams gave a lower prioritization based on this argument. The two other teams discussed it but did not take it into account in their decision.

*Intensivist: It could well be that the guy with Down syndrome, who won't take instructions as easily, will take much longer to eventually get off that ventilator.* (Hospital 4, session B)

*Unease at applying non-medical criteria of the guidelines.* Several physicians expressed unease at the idea of having to apply the non-medical criteria, specifically comparing patients based on age category and drawing lots. One physician said, reflecting on the triage session he was a part of:

*Intensivist: Drawings lots, well, we hadn't done that in our group at all while it was used in other groups. So we apparently tried to make some sort of medical reflection to avoid using the lottery system.* (Focus group 1)

Others, on the other hand, did not feel triage teams should always make decisions based on medical arguments when there is uncertainty:

*Non-intensivist: I also think it is interesting that it is sometimes said, coming from within different parts of the organization*: "*We want to avoid using the lottery system at all times.*" *I don't rate us that highly as medical professionals. We can have an idea about a prognosis, but we are also very often wrong*: "*Does one patient really have a better prognosis than the other*?" *And if we're not sure about that, well, then I have no problem drawing lots and saying*: "*We actually don't really know.*" *(Focus group 3)*

*The team's own considerations: The content.* Most team members expressed arguments that they considered intuitively relevant for triage but that were not, or not clearly, part of the Dutch triage guidelines. Some considerations were used for individual patients, to determine which patients should not be admitted to the ICU at all during phase three irrespective of the current availability of beds. Other arguments were additional considerations when prioritizing

multiple patients. Sometimes, these considerations clearly played a role in the decisions made, while at other times they did not, or it was less clear whether they played a role.

Considerations for individual patients—Most teams considered the situation of individual patients more strictly than the (fictional) referring intensivist. One consideration was whether ICU admission was *medically necessary and beneficial* for an individual patient. For example, one team deemed the breathing problems of the patient with the suicide attempt (R1/R5) not urgent enough to warrant ICU admission, especially during a crisis. Furthermore, many teams discussed whether it could truly be assumed it was *the patient's wish to be admitted*, for example in light of their expected low quality of life after ICU discharge. The need to have a "good conversation" with the patient before referring them to ICU triage was frequently mentioned.

Current mental state—One intensivist mentioned that the extent to which a patient wants to fight to survive could play a role when prioritizing patients for ICU admission. Another participant mentioned that someone's *emotional well-being* played a role for them:

> *Non-intensivist*: *Feeling down does play a role when it comes to this, because if we're talking about a very happy HIV-positive patient who says: 'I still enjoy the little things', then it becomes more difficult, but even though that feeling of being down and depressed is not allowed to play a role right now, when it comes to a crisis, it does play a part: what is their outlook on life*? (Hospital 2, session A)

Situation after ICU discharge—Some patients, for instance the patient with Parkinson's disease (C4) were expected to need a long time to *rehabilitate* after ICU discharge, possibly also resulting in a low *quality of life*. Some team members felt that they should be able to let this play a role in their decisions when prioritizing patients.

> *Non-intensivist*: *We're only talking about him spending less time in the ICU. But his rehabilitation process afterwards also becomes a problem, of course. He becomes delirious. If he were to get out of there, then that's when the trouble really starts. [. . .] And, well, that is not allowed to play a role. And I do find that important*. (Hospital 4, session B)

Furthermore, many teams considered *low life expectancy due to psychiatric reasons* a potential argument for lower prioritization. This was mentioned in the context of the suicidal patient in both sessions (R1/R5).

Social context—Social context of patients was sometimes mentioned as potential argument during the triage sessions. However, this was in practice always followed by a statement from the same participant or other team members that this should not play a role in triage. Mentioned are for example behavior that caused or exacerbated the health condition of the patient (attending illegal party and catching COVID-19 (C7), drinking and smoking (R7)), a mother with a child that needs her (R2), a person refusing care previously (C9), and the impact that having a very small social network may have on longer-term prognosis in case of suicidality (R1/R5).

Other—Several teams mentioned the intuitive relevance of *pregnancy* as an argument to prioritize a patient: this would mean saving two lives. Some team members mentioned that the idea of not being allowed to prioritize pregnant women was distressing for them.

> *Non-intensivist 1*: *There are young pregnant women dying because of us.*

> *Non-intensivist 2*: *Terrible really, isn't it*?

*Non intensivist 3*: *Yes, with this system we will let women who have recently given birth, pregnant women, die.*

*Non-intensivist 2*: *But it is also very complicated to draw lines in that.*

*Non-intensivist 3*: *Yes, that's true.*

[. . .]

*Non-intensivist 1*: *If we get something like this and it's for real, I will end up in psychological distress about those pregnant women*! *(Hospital 1, session A)*

Furthermore, several teams expressed unease at patients with COVID automatically being prioritized over patients needing a semi-elective surgery (R7).

*Non-intensivist*: *I do think*: *someone who needs surgery. . . You don't want all the non-COVID patients to become second. . . their situation will worsen, eh*? *That's just like those people in need of cardiac surgery who are waiting right now. They will go into that operating room in a worse condition. So, when it comes to that [acute abdominal aneurysm, case R7], I wonder*: *should we wait until it goes wrong*? (Hospital 4, session B)

*The team's own considerations*: *Their role*. Opinions differed between professionals regarding the extent to which a triage team should be able to use arguments that are not in the guidelines. When directly asked in the focus groups, no one suggested that arguments not included in the guidelines at all may be used. However, in some sessions these arguments did appear to play a role in decision-making. For example, in one of the sessions, the suicidal woman (R1/R5) was, after a long discussion, put in last place, even though when following the guidelines she should be admitted to the first available bed. In another session, decisions were mainly based on the team's own considerations without looking closely at the guidelines, though in practice the majority of the arguments did overlap.

Additionally, some professionals expressed that they felt it was acceptable for the team's own considerations to play a minor role as long as these were "translated" into criteria of the guidelines. For example, estimations of chance of survival are challenging. This uncertainty gives room for other arguments to play a role: the deciding factor for "is there a 20% difference in chance of survival yes or no" may in practice be an argument unrelated to ICU survival, for example expected difficulties with rehabilitation.

*Intensivist 1*: *And do you mean to say that, if chances are similar on paper, but it says*: *one wants to fight, and the other one does not want to rehabilitate endlessly, then we will not draw lots, but we will argue that it is 10% with those words.*

*Intensivist 2*: *Yes, that can determine how, what those percentages look like, I think.*

*Intensivist 1*: *Yes, yes exactly. But then turn that 10% [into 20%] because we are faced with the choice*: *do they differ by 20%, but according to the records and the hard data it does not, but by carefully considering the story you turn it into 20%.* (Focus group 1)

However, others felt that the protocol had to be followed very strictly, without adding one's own interpretations. For some, the fact that the guidelines are legally binding also played a role in these considerations.

*Non-intensivist*: *I feel a little differently about this, because I really do think that you have to follow this process properly because it has such big consequences and I really think you have to stick to the protocol until you reach the right step in this process to say: this patient will be removed from the list.* (Hospital 1, session A)

*Intensivist*: *It's that simple and I also insisted on that during the assessment [the triage]: guys follow the scheme, because you also have to take the legal consequences into consideration. That's how it is.*

*Non-intensivist 1*: *You have no choice but to ensure that you have followed the right steps.*

*Intensivist*: *You have to prove-*

*Non-Intensivist 2*: *You have to look at the numbers.*

*Intensivist*: *And those numbers are written in black and white, so if you can show them later-*

Non-Intensivist 1: It's very annoying when you have emotions involved-

Non-intensivist 2: And however many emotions there are. . .

*Intensivist*: *It doesn't matter.*

*Non-intensivist 2*:*. . .you don't get paid for that and so you follow the steps and then that's too bad.* (Hospital 1, session B)

**Group processes.** *Opinion on the use of guidelines for triage.* Not all professionals believed that the use of triage guidelines would be feasible during an actual crisis. One team in first instance barely used the guidelines, as one team member felt that they were capable of making these decisions themselves, and that this was also how this would take place in reality. However, the other teams members did not all agree on this approach and eventually the guidelines were also consulted to some extent.

*Non-intensivist*: *What I struggle with a lot as well is, are we doing all this as a practice run or are we trying to be more real-life? So when I get this list, I have that priority list made within five minutes and I think: that's how it will go. (Hospital 2, session A)*

*Careful application of the guidelines.* The teams sometimes skipped criteria, or did not apply them exactly as written down, because they were not systematically following the step-by-step process. The teams were not always aware of this. For example, teams sometimes seemed to lose sight of how prognosis can only be used to prioritize one patient over another if there is a difference of more than 20% in expected chance of survival. In practice, the discussion would sometimes stop with one team member stating that the prognosis of patient A was better than that of patient B, and therefore patient A had priority. In some instances other group members would actively question this and encourage the team to stay close to the guidelines, while in other instances they did not. For example, in one session the comparison of prognoses was only briefly mentioned by the intensivist and a quick conclusion was drawn by the team:

*Intensivist*: *No, I think that [case C3] has better chances compared to other COVID patients, superior chances.*

*Non-Intensivist 1: So then she'll come in between. . .*

*Intensivist*: *Between those two young people and-*

*Non-Intensivist 2: And those two older ones. (Hospital 4, session A)*

The difference in chance of survival was not a given for the abovementioned cases: one of the other teams decided they could not be certain that the difference was 20%.

In general, team members often helped and corrected each other, also when it came to applying the criteria in the right order.

*Non-Intensivist 1: And you're right. If you are going to make choices, you should, first of all, look at the patients who according to expectations need a relatively short ICU admission. And after that...*

*Intensivist: Yes. You can't say anything about that for these patients.*

*Non-Intensivist 2: Well first, the initial question is whether they have a comparable chance of survival. And—I think- that's where we are still at right now.*

*Intensivist: Yes, I think that's... Yes. (Hospital 2, session B)*

*Handling of missing information.* Information on the Clinical Frailty Scale (CFS) was missing for some of the cases. For case R4, two teams scored the CFS themselves based on the (limited) information in the patient file and concluded the patient should be excluded. The two other teams did not discuss this possibility.

*Non-intensivist 1: Oh, guys, hey! That [case R4] moved his bed downstairs so he doesn't have to walk up the stairs.*

*Intensivist: Now that he's ill, maybe he's a bit different, but it's about right before he got ill, what is his clinical frailty score.*

*Non-intensivist 1: [clinical frailty score] 6 is: inside the house they often have trouble walking the stairs and they need help showering. That goes in the direction of 5–6, right?*

*Intensivist: That's right. [Case R4], you said?*

*Non-intensivist 1: Yes, [case R4], it says at social aspects/hobbies: he lives with his partner in a farmhouse. They arranged home care for cleaning. That's a 5 or 6.*

*Intensivist: You're right, then he's removed from the list. Good job, [non-intensivist 1]. I even circled it, I did notice it. (Hospital 3, session A)*

*The role of intensivists versus non-intensivists.* The intensivists had the most medical knowledge and expertise to make the medical assessments relevant for triage and therefore most often drew the conclusions for these criteria. This decisive role of the intensivist was both observed by the researchers as well as acknowledged by the focus group participants. For most non-intensivist participants this did not negatively affect how they viewed their role. Several participants expressed that they felt their triage session had been teamwork, even those participants in teams with a dominant intensivist who was in the lead both process-wise and regarding the triage decisions. However, it was generally stressed that it was important that teams have an intensivist who was open to the comments of others.

*Non-intensivist: With us, the intensivist was open to the ideas of others and he certainly took them into account. We also agreed on that when we looked at the legal aspects: "It must be a joint decision!" So that by the end of the process you won't say: "Yeah, but it was your idea."*

*Then you make yourself very vulnerable. [. . .] I would not have liked to get the feeling that "I'm being overruled". Because, of course, that also applies to that frailty scale [guideline steps 1&2]. That's also kind of intuitive, like: "How are the patients doing?" You can also discuss that and then you will have to agree on that together. I don't think it should be a decision taken by one person alone. (Focus group 3)*

*Non-intensivist 1: I enjoyed having a discussion together and seeing how we could complement each other.*

*Non-intensivist 2: I also enjoyed that we do uhm. . . you take the lead and then we take turns, I think it is well-considered. The fact that we know each other well, that you're not afraid to interrupt.*

*Intensivist: There wasn't really a supervisor in the end. We had a good discussion together and we accepted what each of us had to say. (Hospital 3, session A)*

One participant stressed how the guidelines make it possible for all professionals to play a role in the triage process.

*Interviewer: And how did that go then, for example, with the assessment of prognosis and length of stay and that sort of thing?*

*Non-intensivist: It is true that at that point the opinion of the intensivist is more [taken into consideration]—especially when you talk about the differences in prognosis, is it a lot better or a lot worse? [. . .] In all other aspects I think that the flowchart is structured in such a way that you do not necessarily need the expertise for it, or in any case do not, as an intensivist, need to give direction. (Focus group 4)*

However, one physician expressed that they felt it was hard to be an equal partner in the team.

*Non-intensivist: I let myself be guided by the intensivist that was there, because he, of course, had the most experience with the track record of patients in the ICU. And that made it, I sometimes think, difficult to be an equal partner in the team as a non-intensivist. (Focus group 3)*

## Discussion

This study shows that triage teams generally considered the Dutch triage guidelines acceptable and feasible to come to a prioritization of cases. In practice, some teams also allowed their own considerations (outside of the guidelines) to play a role when making triage decisions, for example if this helped to avoid using non-medical criteria such as drawing lots. Group processes also played a role in triage decision-making: triage choices can be influenced by the team members' opinion on the guidelines, the carefulness with which the guidelines are applied by the team and how the team deals with missing information. Intensivists have the most relevant experience for making medical assessments such as a patient's prognosis, meaning they often have the largest role during triage sessions.

Having fair procedures in place that are consistently applied in practice is an important aspect of procedural justice [17, 18]. This study shows that while the Dutch ICU triage guidelines are mostly followed, some inconsistency in medical assessments and therefore in how patients are prioritized seems in practice unavoidable. However, though teams to some extent

used different argumentation, their prioritization of the cases is mostly similar. It appears that the Dutch guidelines contain an inherent consistency: patients with a better chance of surviving the ICU admission are likely to also have a shorter expected length of stay in the ICU, and/or have a lower age. In many cases it is therefore likely that the same patient will be prioritized. In many other countries, the national guidelines describe only the general principles and criteria that triage teams should take into account when making triage decisions, without operationalization of these criteria and also without being legally binding [10]. During a crisis, the guidelines will have to be operationalized locally or by individual teams. It is likely that the procedures followed and the prioritization of patients will have a higher level of inconsistency than was found in this study. This study shows that using a step-by-step protocol is feasible and considered generally acceptable by triage teams, and we therefore hope this study will encourage other countries to take steps to operationalize their triage guidelines.

## Inconsistencies in the application of the Dutch ICU triage guidelines

There were some notable inconsistencies among teams in their application of the guidelines and the triage process. First, some teams took into account their own considerations when prioritizing patients, for example regarding quality of life, in the form of arguments about rehabilitation or emotional well-being. While quality of life is not part of the Dutch guidelines, it is mentioned as (non-operationalized) criterion in the guidelines of other countries such as Belgium, Australia and New Zealand [10]. In the international literature, however, the ethical and practical desirability of using quality of life as a criterion is debated: estimating it is difficult and may incur bias, and it also suggests that the life of a patient with an illness or disability is worth less than that of others [19–23]. An approach might be preferable in which, prior to referral for triage, the wishes of the patient and/or their family are discussed in light of expected future quality of life [9, 24]. Offering training to triage teams in which both the triage criteria and the rationale underlying them are explained may make triage teams less likely to take into account their own considerations.

Secondly, another inconsistency found in this study was how length of stay in the ICU was taken into account for the patient with Down syndrome, in whom prolonged weaning was expected due to cognitive problems. In two teams this argument impacted the prioritization. However, the Dutch guidelines [15, 16], as well as the guidelines of many other countries and international literature [10, 19, 25–28], argue that patients should be considered equally in triage, irrespective of potential disabilities that do not impact chance of short term survival. While taking time to wean from the ventilator technically falls within the scope of the Dutch ICU triage guidelines, this consideration is at odds with the principle of equality as described in the guidelines. Therefore, guideline writers should consider whether cognitive abilities can potentially play a role in their suggested triage criteria and explain whether and how teams are allowed to consider this in their triage decisions.

Thirdly, several triage teams overruled the (fictional) intensivist that had referred the patients for triage, while other teams were unsure if this was allowed. In reality this situation will likely not happen very frequently, because the teams considered some cases unlikely to be referred for triage in their hospital in the first place. For example, patients of which it is doubtful that it is really their wish to be admitted to the ICU. In reality, hospitals are free to exclude patients at the stage of referral for triage: the step-by-step criteria only become relevant once a patient is referred. Furthermore, the Dutch guidelines allow intensivists to apply the in- and exclusion criteria (steps 1 and 2) without a triage team. This means triage teams would in reality likely not frequently be involved in inclusion choices.

Lastly, the participants of this study were often not very familiar with the Dutch ICU triage guidelines, despite them being part of official triage teams and having received some training. The study also showed that the guidelines were sometimes not carefully applied, which teams did not always seem to notice. Therefore, our results stress the need for more extensive training in the use of triage guidelines (for example through fictional sessions) if the guidelines are to be applied in a consistent and precise way. This will be even more relevant in real triage situations in which there is less available time to study the details of the guidelines during the triage session, and team members will experience more stress. Digital tools which guide teams through the triage process step-by-step may also potentially be beneficial.

## Empirical research on prioritization

Several quantitative empirical survey studies have been conducted in which participants (laypersons or nursing students) were asked to choose whom to give care between two patients with different characteristics [29–32]. Patients were often prioritized based on utilitarian principles (maximizing benefit), which is in line with the main principles of international guidelines for ICU triage. However, the studies also show that patients who have an instrumental value to society (e.g. nurses), or who have young children, are more likely to be prioritized [30, 31]; while patients who are perceived to be partially responsible for their health situation (e.g. a patient with severe COVID-19 and obesity, or who did not comply with COVID-19 policy measures) are less likely to be prioritized [30, 32]. We did not observe a role of these types of "context" arguments in decision-making in the current study. When these arguments were occasionally brought up, participants immediately mentioned that the guidelines prohibit taking these factors into account. Even though healthcare professionals may already be more aware of the need to disregard context factors, stressing this in the guidelines therefore does seem an important reminder.

## Morally injurious events

This study found that conducting ICU triage may be distressing for the triage team members. Decisions that will result in patients not receiving life-saving treatment can be considered 'potentially morally injurious events': situations in which moral views and expectations are under pressure [33–35]. It is therefore important that, if this crisis situation becomes reality, suitable psychological support is offered to triage team members.

## Strengths & limitations

The main strength of this study is that it is a thorough empirical study into the application of guidelines, in which real triage teams prioritized cases designed by their peers. The participants were also asked to reflect on these sessions in focus groups to obtain richer data. The study took place in a period in which ICU triage might have become necessary in the Netherlands, resulting in highly motivated teams and rich results.

This study also has several limitations. First, the study took place in one country and with one set of national triage guidelines, limiting generalizability. Second, no data saturation was reached: potentially more factors and arguments would have been found if more sessions had been held with different teams and with different cases. Third, the cases in session B were not as diverse as we had aimed for: many did not meet the inclusion criteria, which limited decision-making in which patients were compared. Fourth, in reality triage sessions are likely much more stressful, considering time pressure and real patient lives being at stake, which may affect decision-making. Fifth, in reality patients with and without COVID-19 are admitted to separate ICU units within each hospital and therefore they would not directly compete

for a bed, as was the case in this study. Lastly, the process of a team coming to a triage decision is to some extent a black box: it is not always clear which, also potentially implicit or unvoiced, arguments or feelings play a role. However, we believe we have gathered enough evidence to conclude that the factors that we found play a role in the process of coming to triage decisions, and may play a role in future triage sessions.

## Conclusions

The Dutch ICU triage guidelines were generally considered acceptable and applicable, and were the main basis for triage decisions. However, considerations not in the guidelines sometimes also played a role when making triage decisions, for example if this helped to avoid using non-medical criteria such as drawing lots. ICU triage guideline writers should consider which aspects of their criteria might, when applied in practice, lead to inconsistencies or ethically questionable prioritization of patients. Offering triage team members training in which the reasoning for the criteria is explained, and in which they can practice applying the guidelines, might improve both the willingness and ability of triage teams to follow the guidelines closely.

## Acknowledgments

We thank the participants of this study for their generous participation and dr. Eva Verkerk for her useful comments on the manuscript.

## Author Contributions

**Conceptualization:** Gert J. Olthuis, Anke J. M. Oerlemans.

**Formal analysis:** Inger L. Abma, Gert J. Olthuis, Irma T. H. M. Maassen, Marjan L. Knippenberg, Anke J. M. Oerlemans.

**Funding acquisition:** Gert J. Olthuis, Annie J. Hasker, Bernard G. Fikkers, Anke J. M. Oerlemans.

**Investigation:** Inger L. Abma, Gert J. Olthuis, Irma T. H. M. Maassen, Marjan L. Knippenberg, Anke J. M. Oerlemans.

**Methodology:** Inger L. Abma, Gert J. Olthuis, Anke J. M. Oerlemans.

**Project administration:** Inger L. Abma, Irma T. H. M. Maassen, Marjan L. Knippenberg.

**Resources:** Miriam Moviat, Annie J. Hasker, A. G. Buenen, Bernard G. Fikkers.

**Supervision:** Gert J. Olthuis, Anke J. M. Oerlemans.

**Writing – original draft:** Inger L. Abma.

**Writing – review & editing:** Gert J. Olthuis, Irma T. H. M. Maassen, Marjan L. Knippenberg, Miriam Moviat, Annie J. Hasker, A. G. Buenen, Bernard G. Fikkers, Anke J. M. Oerlemans.

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
