## [Decision Letter · Decision Letter 0]

24 Apr 2023

PONE-D-22-16565Putting ICU triage guidelines into practice: a simulation study using observations and interviewsPLOS ONE

Dear Dr. Abma,

Thank you for submitting your manuscript to PLOS ONE. After careful consideration, we feel that it has merit but does not fully meet PLOS ONE’s publication criteria as it currently stands. Therefore, we invite you to submit a revised version of the manuscript that addresses the points raised during the review process.

We look forward to receiving your revised manuscript.

Kind regards,

Jianhong Zhou

Staff Editor

PLOS ONE

Journal Requirements:

Reviewers' comments:

Reviewer's Responses to Questions

**Comments to the Author**

1. Is the manuscript technically sound, and do the data support the conclusions?

Reviewer #1: Yes

Reviewer #2: Yes

2. Has the statistical analysis been performed appropriately and rigorously? 

Reviewer #1: Yes

Reviewer #2: Yes

3. Have the authors made all data underlying the findings in their manuscript fully available?

Reviewer #1: Yes

Reviewer #2: Yes

4. Is the manuscript presented in an intelligible fashion and written in standard English?

Reviewer #1: Yes

Reviewer #2: Yes

5. Review Comments to the Author

Reviewer #1: The authors have addressed a very important topic in their research. The article is written in an eye catching manner and are the methodological details have been provided. The results are presented appropriately.

Reviewer #2: The topic is very interesting & practical testing of triage guidelines in a scientific study has so far not

taken place. The study findings would be helpful for hospitals adapting national guidelines. I have few comments on the study:

Well-developed methodology is there but results are bit confusing, and the numbering mentioned in table 5 are not understandable. More brief points for this should be mentioned.

Try to elaborate introduction and discussion part as they are not properly depicting the main theme and results of your done work.

6. PLOS authors have the option to publish the peer review history of their article (what does this mean?). If published, this will include your full peer review and any attached files.

Reviewer #1: No

Reviewer #2: No

---

## [Author Response · Author response to Decision Letter 0]

8 May 2023

We thank the editor and the reviewers for their helpful comments. Below, a point by point response is provided.

>> We have made the required adjustments. It should be noted that we have four levels of headings, but we saw no way to reduce this to three levels without making our (rather complicated) results more confusing. We have for now used italics for this fourth level (but can adjust this if needed). We have also used italics for the quotes in the text, so that they are not confused with the main text; as well as for certain phrases that correspond with the main topics that we found to provide more structure in the text. These can be removed if this is desired.

We also noted that the numbering for the tables and footnotes in the tables was not always correct. This was also adjusted.

>> We have added the information that the participants signed written informed consent. The study did not include minors.

>> The reference list was checked and was found to be complete and without retracted articles. We made some adjustments to the layout of some of the references and added hyperlinks to (non-article) documents such as guidelines. 

Reviewer #1: The authors have addressed a very important topic in their research. The article is written in an eye catching manner and are the methodological details have been provided. The results are presented appropriately.

Reviewer #2: The topic is very interesting & practical testing of triage guidelines in a scientific study has so far not taken place. The study findings would be helpful for hospitals adapting national guidelines. 

>> We thank the reviewers for their appreciative comments.

I have few comments on the study: Well-developed methodology is there but results are bit confusing, and the numbering mentioned in table 5 are not understandable. More brief points for this should be mentioned.

>> We agree that Table 5 is a bit of a puzzle. We only noted down the codes of the cases (rather than a description) because otherwise the table becomes very cluttered. This means the codes have to be explained outside the table. Rather than using footnotes to repeat the information in Table 3 (in which the codes are explained), we have a footnote referring to Table 3. We still believe this is the most elegant way to do it, as the footnotes section is already very long for this table. However, to better aid readers in understanding Table 5 we have added the following:

• When referring to Table 5 in the text we directly encourage readers to use Table 3 to understand the codes. P11: Prioritization of patients was similar among the teams, though not exactly the same (Table 5; see Table 3 for case codes).

• We have redesigned Table 3 in order to have the codes of the cases stand out more (P9). This should make it easier to look up which case each codes refers to. Additionally, we have made explicit in the title of Table 3 that this table explains the coding system used in the article.

We hope that these changes will make Table 5 easier to understand. 

Additionally, while doing this, we realized that one footnote/explanation was missing from the table. We added this footnote and adjusted the letters of the other footnotes accordingly (the footnotes are also changed from numbers to letters, as per the journal’s style requirements).

Try to elaborate introduction and discussion part as they are not properly depicting the main theme and results of your done work.

>> We agree that the introduction section should be more elaborate. We have added a paragraph with general information regarding ICU care and COVID-19 patients in the ICU, which we feel will help readers understand the context of our research better (page 1).

Regarding the discussion section (currently 1500+ words): after discussion in the research team we remain unsure how we could better depict the main theme and results of our work, as per the reviewer’s request. Currently the discussion section covers our most important findings, compared to international guidelines and literature, and with their practical implications (i.e. need for operationalized guidelines; need for more training of triage teams; need for psychological support for triage teams); as well as a comparison with literature on prioritization in healthcare in general. 

Unfortunately, we have therefore not been able to add new information to the discussion section based on the reviewer comment. We hope the reviewer will reconsider this aspect of their comment or otherwise share which topics they feel are missing from the discussion section.

---

## [Editor Report · Decision Letter 1]

29 May 2023

Putting ICU triage guidelines into practice: a simulation study using observations and interviews

PONE-D-22-16565R1

Dear Dr. Abma,

We’re pleased to inform you that your manuscript has been judged scientifically suitable for publication and will be formally accepted for publication once it meets all outstanding technical requirements.

Kind regards,

Amjad Khan, Ph.D.

Academic Editor

PLOS ONE
---

## [Editor Report · Acceptance letter]

15 Aug 2023

PONE-D-22-16565R1 

Putting ICU triage guidelines into practice: a simulation study using observations and interviews 

Dear Dr. Abma:

I'm pleased to inform you that your manuscript has been deemed suitable for publication in PLOS ONE. Congratulations! Your manuscript is now with our production department. 

Kind regards, 

on behalf of

Dr. Amjad Khan 

Academic Editor

PLOS ONE